# The Severity of Pain in Prostate Biopsy Depends on the Biopsy Sector

**DOI:** 10.3390/jpm13030431

**Published:** 2023-02-27

**Authors:** Grzegorz Rempega, Paweł Rajwa, Michał Kępiński, Jakub Ryszawy, Jakub Wojnarowicz, Maksymilian Kowalik, Marcela Krzempek, Aleksandra Krzywon, Michał Dobrakowski, Andrzej Paradysz, Piotr Bryniarski

**Affiliations:** 1Department of Urology, Division of Medical Sciences in Zabrze, Medical University of Silesia in Katowice, 41-800 Zabrze, Poland; 2Department of Urology, Medical University of Vienna, 1090 Vienna, Austria; 3Department of Biostatistics and Bioinformatics, Maria Skłodowska-Curie National Research Institute of Oncology, Gliwice Branch, 44-100 Gliwice, Poland; 4Department of Biochemistry, Faculty of Medical Sciences in Zabrze, Medical University of Silesia in Katowice, ul. Jordana 19, 41-808 Zabrze, Poland; 5Department of Radiology and Radiodiagnostics, Public Clinical Hospital, Medical University of Silesia in Katowice, ul. 3-go Maja 13-15, 41-800 Zabrze, Poland

**Keywords:** prostate biopsy, pain, prostate cancer

## Abstract

BACKGROUND: The pain experienced by a patient during a prostate fusion biopsy is cumulative and can also be modulated by many factors. The aim of the study was to assess the association between the degree of pain intensity during prostate biopsy and the region of the biopted organ. MATERIALS AND METHODS: The study included a group of 143 patients who underwent prostate fusion biopsy under local analgesia followed by blockage of the periprostatic nerve. After a biopsy, the patients completed the original questionnaire about the pain experienced during the procedure. RESULTS: There was a statistically significant difference in pain score between cores taken in the apex (median 5 (IQR 2–5)), medium level (median 1 (IQR 1–2)), and prostate base (median 1 (IQR 1–3)) (*p* < 0.001). The malignancy scale ISUP ≥ 2 (*p* = 0.038) and lower PSA value (r = −0.17; *p* = 0.046) are associated with higher pain during procedure. Biopsy time was correlated with discomfort (r = 0.19; *p* = 0.04). Age (*p* = 0.65), lesion size (*p* = 0.29), PI-RADS score (*p* = 0.86), prostate volume (*p* = 0.22), and the number of cores (*p* = 0.56) did not correspond to the pain scale. CONCLUSIONS: The apex is the most sensitive sector of the prostate. ISUP ≥ 2 and patients with low PSA levels more often indicated higher values on the pain rating scale.

## 1. Introduction

Prostate cancer (PCa) is one of the most commonly diagnosed malignant neoplasms in the world. It ranks second (after lung cancer) with over 1.4 million patients [1,2]. In the initial stages, it is asymptomatic, and the first symptoms appear only in more advanced stages. These symptoms (urinary disorders, pollakiuria, and pain in the lower abdomen) are non-specific and may be confused with prostate hyperplasia and ongoing inflammation. The factors predisposing to the disease are age (the probability increases with the age of the patient), black ethnicity, hypertension, dietary factors and obesity. Due to the fact that ethnicity and positive family history are associated with an increased incidence of PCa, a genetic predisposition is suggested. The high-risk group includes men over 50 years of age, and in the case of a positive family history of PCa, over 45 years of age [3,4].

Most cancers are detected in the asymptomatic phase, which is possible thanks to the widely developed prophylaxis including the prostate-specific antigen (PSA) blood level test. Digital rectal examination (DRE) is also helpful [3].

The method of treating prostate cancer depends on many factors. These factors include: the patient’s general health (including comorbidities), age, and tumor stage. The analysis of these factors allows the decision on surgical treatment, radiotherapy, systemic treatment, and active surveillance to be made. It is mainly patients with localized cancer that qualify for local treatment (surgery or radiotherapy) [5].

PCa diagnostics is based on imaging tests such as transrectal ultrasound (TRUS) or multi-parametric magnetic resonance (mpMRI). The Prostate Imaging and Reporting and Data System (PI-RADS) scoring system is used to evaluate the obtained images in mpMRI and to prepare descriptions [6]. Imaging methods are necessary to assess the location of the tumor or to precisely perform a targeted biopsy, which is a necessary examination in the diagnostics of PCa [7,8,9,10].

Fusion biopsy allows for the histological assessment of neoplastic tissue using the Gleason score scale, which is necessary to predict treatment results and assess the prognosis [11]. The most recent biopsy techniques include a transperineal approach under local analgesia [12]. The fusion method is more sensitive and specific than a systematic biopsy. It allows one to avoid the need to repeat the biopsy in the case of a suspected false-negative result obtained in a systematic biopsy. It is estimated that approximately 50% of patients who were negative in traditional biopsy were later diagnosed with PCa [10,12,13].

The gold standard in prostate analgesia before biopsy is local analgesia with periprostatic nerve bundle block (PPNB). It is an easy, safe and highly effective method. In the case of young men, as well as among patients with prostate hyperplasia, the periprostatic nerve block itself is not sufficient, and the reduction of pain sensation is possible by using additionally perianal and rectal gel with lidocaine [14,15,16].

The pain threshold is individual and subjective, and therefore it is difficult to clearly define or estimate. The pain experienced by a patient during a prostate biopsy is cumulative, which means that an increase in the number of punctures directly correlates with an increase in total pain. It can also be compounded by shame and stress before the digital rectal examination, stress before the biopsy itself and fear of sexual dysfunction or diagnosis. Additionally, physical factors such as the size of the prostate and the age of the patient have an impact on the level of pain experienced. Literature data indicate that younger patients experience stronger pain compared with older patients, which is associated with greater tension of the anal sphincter muscle [15]. The intensification of pain sensations may also be caused by activities performed before the biopsy, i.e., digital rectal examination, or insertion of an ultrasound head or a needle for injecting analgesics through the rectum [16]. Appropriate scales are used to assess the level of perceived pain. The most commonly used is the visual analog scale (VAS). It is a 10-point scale where 0 points means no pain at all and 10 points means pain unbearable for the patient.

Despite the widespread use of fusion biopsy in cancer diagnostics, the aspect of pain during the procedure still leaves many unknowns. The available literature indicates numerous inconsistencies on this topic. Therefore, in our study, an attempt was made to assess the degree of pain experienced by the patient during the fusion biopsy and to standardize the information on this issue.

The aim of the study was to assess the difference between the degree of pain intensity during prostate biopsy and the region of the biopted organ. Additionally, our study confirms the hypothesis that the apical and anterior areas of the prostate are more sensitive to pain. We also examined how factors such as age, lesion size, prostate volume, number of samples taken, PIRADS, ISUP and PSA values affect the level of pain experienced by the patient during prostate fusion biopsy.

## 2. Material and Methods

### 2.1. Study Design

The study included a group of patients with suspected prostate cancer who underwent prostate fusion biopsy from June 2018 to June 2022 at the Department of Urology, Clinical Hospital number 1 in Zabrze. The biopsy was performed under local analgesia with transrectal access. The procedure was performed using ultrasound with a BK Medical Flex Focus 400-BioJet System (DK Technologies, (Barum, Germany). Before the biopsy 11 mL of Lignocaine gel was administered to the rectum, then, around the periprostatic nerve bundle, 5 mL of 1% lignocaine was injected on each side. Analgesia was performed with a Chiba 18 G needle. Samples were taken by The Pro-Mag Ultra biopsy gun with an AIOU 14 G 250 mm needle. Initially, 4 samples were taken from a suspicious lesion in MRI, and then 8–12 samples were systematically taken from the right and left lobes of the prostate. The number of samples depended on the assessment of the quality of the collected material by the operator during the biopsy. If the collected material was not suitable for histopathological evaluation, more specimens were collected. Furthermore, though the MRI fusion biopsies were performed by two trained and experienced urologists, there could nevertheless be some impacts from factors at the physician level on the analyzed outcomes. Before collecting the material for histopathological examination, each patient was interviewed to garner information on their age, weight, height, PSA value, DRE result, nicotine addiction, family history in terms of PCa, the result of any previous biopsies, prostatitis or diseases, and concomitant and previous pelvic surgeries (Appendix A). After biopsy, the patients were asked to complete the original questionnaire (Appendix A), which was aimed at the subjective assessment of the patient about the pain experienced during the administration of analgesia, insertion of the ultrasound head, maneuvering and sampling. The perceived pain was assessed using a ten-point visual analog scale (VAS), 5 min after the procedure. The questionnaire also included the level of stress, perceived discomfort and a declaration as to whether the patient would undergo another biopsy if it was necessary to repeat it (tolerance). For the statistical evaluation of the most sensitive point of the prostate, the sectors from which the biopsies were collected were numbered 1–18 (the sector numbering was original) (Appendix A), and then compared with the level of pain experienced by the patients.

In the course of the analysis, we examined how factors, such as the PSA value, PIRADS, or the previously performed biopsy, influence anxiety during the procedure. We also assessed whether the prolonged duration of the biopsy correlates with the perceived discomfort, pain and possible development of tolerance. The study also focused on the analysis of whether age, lesion size, prostate volume, number of samples taken and PIRADS, ISUP and PSA values have an impact on the level of perceived pain.

Overall, we excluded patients who had neurological disorders or history of neuropathy, had received anticoagulants, or had no evaluable MRI reports.

### 2.2. Statistical Analysis

Categorical variables were summarized as frequencies and percentages. Continuous and ordinal data were shown as median values with interquartile ranges (25 to 75%, IQR 25–75) and as mean values with standard deviation ranges as well with min/max ranges, unless otherwise stated. Differences between two groups were determined using Wilcoxon rank sum test (aka Mann–Whitney U test) and for more than two groups, comparisons were performed by Kruskal–Wallis H test. Spearman correlation coefficient was assessed to examine the correlation between variables. The classification of the correlations used to evaluate the analyzed results was as follows: 0.0 ≤ |r| < 0.1 negligible correlation, 0.1 ≤ |r| ≤ 0.39 weak correlation, 0.4 ≤ |r| ≤ 0.69 moderate correlation, 0.7 ≤ |r| ≤ 0.89 strong correlation, 0.9 ≤ r ≤ 1 very strong high correlation [17]. A two-sided *p*-value < 0.05 was considered statistically significant, and *p*-value < 0.10 was considered close to statistical significance. All analyses were performed using R software package version 4.0.1 released on 6 June 2020 (R Foundation for Statistical Computing, Vienna, Austria, http://www.r-project.org, accessed on 31 October 2022).

## 3. Results

In total we included 143 patients aged 41 to 84 years who underwent transrectal fusion prostate biopsies. Of these, 69 (48%) were biopsy naïve, and 74 (52%) had a history of previous prostate biopsy (1–5 biopsies). The median age was 66.4 (IQR 61.2–70.8) years, the median PSA was 7.60 (IQR 5.91–10.11), median volume TRUS (ml) was 50 (IQR 35–63), and median PSAD (PSA/prostate volume) was 0.16 (0.11–0.26) (Table 1). Ninety-five patients (66%) had nonsuspicious DRE and 134 (93%) were diagnosed with PI-RADSv2 ≥ 3 on MRI with median target lesion diameter of 14 (IQR 9–16) mm.

Notably, targeted lesions were located in apex in 24% of patients, in mid in 46%, and base in 30%. In 57% of patients targeted lesions were located in the posterior zone of the prostate and in 43% in the anterior zone of the prostate. Following collection of median 14 cores (IQR 12–16), 29% of patients had ISUP GG ≥ 2. The mean biopsy time was 31 min. Most biopsies were localized in sector 8 (22 out of 143 performed biopsies), which accounted for 15% of the respondents (Figure 1). None of the targeted biopsies involved sectors 1, 4, and 17.

The maximum pain score (0–10 scale) for separate parts of fusion biopsy were as follows: USG insertion 5, periprostatic block 6, DRE 6, ultrasound manipulation 5, sample cores collection 8. Of note, only 11% of patients reported zero pain for core collection. Most of the patients reported some level of pain, however none indicated a maximum score of 9–10. Overall, 56% reported a good tolerance of fusion biopsy (score 0–1). With regard to tolerance, the median pain score for the entire study group was 1 (IQR 0–2). Comparison of the development of tolerance among the subjects by subgroup (apex (*n* = 34; median 1 (IQR 0–2.75)); medium level (*n* = 66; median 1 [IQR 0–2]); base (*n* = 43; median 1 (IQR 0–2))) did not show significant differences (*p* = 0.97). Forty-five percent reported a low level of anxiety (score 0–1).

We found some evidence for higher median pain scores for cores located in the posterior zone compared with the anterior zone of the prostate, however it did not reach a statistical level of significance (2 (IQR 1–4) vs. 1 (IQR 1–2), *p* = 0.099). On the other hand, there were statistically significant differences in pain score between cores taken in the apex (median 5 (IQR 2–5), medium level (median 1 (IQR 1–2)) and prostate base (median 1 (IQR 1–3)) (*p* < 0.001) (Figure 2, Table 2). Considering clinical data, there was no difference between anxiety before the biopsy and PSA values (*p* = 0.84), PI-RADS (*p* = 0.47) or history of prostate biopsies (*p* = 0.49). Increasing biopsy duration was correlated with increased discomfort (r = 0.19, *p* = 0.04).

We further tested if established PCa risk factors, including age, MRI lesion diameter, PI-RADS, ISUP GG, PSA, number of cores and prostate volume are associated with our biopsy pain score (Table 3). Overall, age (*p* = 0.65), lesion size (*p* = 0.29), PI-RADS score (*p* = 0.86), prostate volume (*p* = 0.22), and, importantly, number of cores (*p* = 0.56) did not correspond to pain scale.

Patients with higher PSA value more often indicate lower pain score (*p* = 0.046). However, the analysis of medians and IQR, despite the visible slight trend, did not show statistically significant differences (*p* = 0.16) between the studied groups (Figure 3, Table 2).

The result of the ISUP GG analysis clearly indicates statistically significant differences between patients with ISUP 0–1 and ISUP 2–5 ((IQR 1–5); *p* = 0.038) (Figure 4, Table 2). Patients with ISUP 2–5 more often indicate higher pain scores.

Smaller prostate volume was also correlated with higher pain during rectum examination (r = −0.28, *p* = 0.002). Moreover, older patients relatively more often indicated lower values of the VAS pain scale compared to younger patients, however, this result was not statistically significant (*p* = 0.18).

## 4. Discussion

The main aim of the study was to assess the difference between pain intensity during prostate fusion biopsy and the region of the biopted organ. The obtained results allowed for the confirmation of the research hypothesis and made it possible to achieve additional goals of the conducted analysis.

Thanks to the use of local analgesia and blockade of the periprostatic nerve bundle, prostate fusion biopsy is not a painful procedure. None of the examined patients rated the accompanying pain at 9–10 points (on a ten-point VAS scale). Only one person reported level 8 of perceived pain during the sampling stage (which accounted for 0.7% of the respondents), and 11% indicated the lack of pain at this stage of the study. The remaining stages of the procedure, such as administration of analgesia, DRE, insertion of the ultrasound transducer, or manipulation of the ultrasound transducer were assessed at a maximum of 5–6 points. The median tolerance during the fusion biopsy procedure was 1.

Similar results were also presented in the available literature, where the fusion biopsy, in the assessment of patients, was associated mostly only with the feeling of acceptable discomfort. Already in the 1990s, scientists showed that 95% of the subjects felt only slight discomfort during the insertion or manipulation of the ultrasound head. Ninety-two percent of patients felt discomfort during biopsy. A similar effect was obtained during research conducted at the Medical University of Marseille, where 131 patients who underwent prostate fusion biopsy without analgesia were analyzed. The vast majority of respondents assessed the procedure as painless, with acceptable discomfort. In turn, Bosnian scientists conducted a similar study on a group of 90 patients, dividing them into three subgroups: The first received a blockade of the periprostatic nerve, the second received Voltaren suppositories, and the third underwent an examination without analgesia. The average pain rating (on a ten-point scale) in group 3 was 6.06 +/− 2.95, which turned out to be statistically significant (*p* < 0.001). For comparison, the subjects included in group 1 described the fusion biopsy as a painless procedure, assessing it at 3.10 +/− 2.32 on the pain scale. Additionally, patients who did not receive analgesia classified it as painless [18,19]. The use of analgesia can reduce pain and discomfort during the fusion biopsy; however, it should be taken into account that each patient may respond differently to the type of analgesia used [20,21].

It was also examined whether the duration of the biopsy correlated with the perceived discomfort, the level of pain and the possible development of tolerance. The effect on the perceived discomfort (*p* = 0.038), which increased with the increasing duration of the procedure, turned out to be statistically significant. A similar study was conducted in Japan, but only the correlation between the duration of the procedure and the pain experienced was analyzed. Pain was assessed at each puncture (using the faces pain scale) and immediately after withdrawing the transrectal probe. There was a clear correlation between the duration of the biopsy and the pain experienced [22].

The obtained results of the statistical analysis assessing the difference between the region of the sample taken and the level of perceived pain clearly indicate that the most sensitive sector is the peak part (*p* < 0.001). The apex of the prostate is an extremely pain-sensitive site during biopsy due to the predominant presence of somatic nerves in the area below the dentate line. The result obtained by us coincides with the assumption of our work and conclusions published in the available literature [15,23,24,25]. Moreover, French researchers have shown that the first prick during a biopsy is the most painful for the patient. Therefore, they concluded that, because of the greatest sensitivity of the apical part of the prostate, it is recommended to start collecting biopsies from the base sector of prostate [23].

Increased sensitivity of the apical part of the prostate was also demonstrated by scientists from Yonsei University College of Medicine, who divided the 312 participants into two subgroups (subgroup 1—periprostatic nerve block of the base; subgroup 2—periprostatic nerve block of the base and apex). The pain score increased from base to apex (*p* < 0.001) and pain increased with the duration of the procedure. In the second group, the level of perceived pain was low and did not increase with time [26]. The same study was repeated on a group of 229 consecutive patients. Pain was assessed at the time of insertion of the probe, injection at the base of the prostate, injection at the tip of the prostate, taking samples, and 15 min after biopsy. The analysis showed stronger pain among patients in group 1 [27]. These results confirm the increased sensitivity of the apical part of the prostate and are consistent with the results obtained in our study.

Additionally, we assessed which of the prostate zones (anterior/posterior) is characterized by increased sensitivity. Unfortunately, the obtained result does not confirm the earlier assumption that the anterior biopsy is more painful (*p* = 0.099). This hypothesis was dictated by the difficult access to this part of the organ [28]. The lack of difference may be due to the subjective assessment of the degree of pain experienced by the patients, taking into account the fact that pain sensitivity is an individual feature. It should also be taken into account that the procedure was performed under analgesia, and that each patient could respond to it with a different severity.

In the course of the analysis, we examined whether the earlier undergoing of a biopsy and the awareness of the PSA and PIRADS results had an impact on the anxiety experienced before the procedure. None of the above-mentioned factors turned out to be statistically significant. However, the literature data indicate the presence of a positive correlation between the declared previous biopsy and pain during the subsequent procedure [15,29,30]. It is possible that the different result of our study is related to the subjective, difficult to estimate assessment of pain experienced by the patient and the reluctance of the surveyed men to admit their fear before the procedure. It has been shown that about 50% of men associate the necessity of undergoing a biopsy with anxiety and severe stress, which suggests that the procedure itself is a heavy mental burden. Anxiety is believed to cause an exaggerated perception of pain [15,29]. The information contained in the literature shows a positive correlation between the anxiety experienced by the patient before the start of the procedure and the pain experienced during its duration [22]. The fear of a planned biopsy is also exacerbated by a longer waiting time for the procedure, which results in increased discomfort during its performance [31]. Unfortunately, the result obtained in our study does not show any correlation between the anxiety experienced before the procedure and a higher assessment of the perceived pain (*p* = 0.38).

The analysis of the influence of the patient’s age on the level of perceived pain, did not indicate a correlation between these factors (*p* = 0.65). Initially, it was assumed that younger age would be associated with greater pain, as younger patients were characterized by greater anal sphincter muscle tone [15,32]. Apart from difficulties in estimating the perceived pain, another aspect that could have influenced the obtained result is the fact that the questionnaire was completed after the end of the examination, regardless of its duration and the number of samples taken. It is possible that the procedure is shorter for younger patients than it is for elder patients, and that this was associated with a comparable level of pain perception.

Another factor that we examined was the influence of the size of the lesion on pain. The result obtained (*p* = 0.29) indicates no correlation. A different result was obtained by French and Korean researchers, who showed that the size of the neoplastic lesion is significantly related to the level of perceived pain [15,23]. The discrepancy in the results obtained may be associated with the limitations of our work.

The correlation between the PSA value and the level of pain experienced by the patient was also analyzed. Initially, it was noted that patients with higher PSA values were more likely to indicate lower pain scores (*p* = 0.046). A similar trend has also been demonstrated in the available literature [15,23]. However, further statistical analysis of the medians and IQR, despite the apparent slight trend, did not show statistically significant differences (*p* = 0.16) between the study groups. This result indicates that it would be worth repeating the study (without limitations) in terms of assessing the impact of PSA value on pain experienced during biopsy.

A visible relationship was found between the ISUP value and the intensity of pain associated with prostate fusion biopsy. Patients with ISUP ≥ 2 indicated more severe pain in comparison with patients with ISUP 0–1 (*p* = 0.038). This result is relatively pioneering due to the lack of access to similar studies. For this reason, it is worthwhile to design a retest focused solely on the correlation between ISUP 2–5 and biopsy pain with more test subjects and thereby eliminate most of the limitations of our work.

The group of patients we studied was diversified in terms of the number of collected biopsies. This allowed for the finding of no difference between the number of samples and the pain associated with the biopsy (*p* = 0.74). The previously mentioned analysis by Japanese researchers also excludes this relationship [22]. On the other hand, studies carried out in Korea and Israel showed the intensification of pain associated with the increased amount of samples taken during the biopsy [33,34]. The increased number of samples taken is also associated with the extended duration of the procedure, which may make it difficult to isolate a significant factor.

Initially, it was assumed that the increased volume of the prostate, due to the need to collect more biopsy samples and the prolonged duration of the procedure, would positively correlate with increased pain sensation. The obtained result (*p* = 0.47) does not allow us to confirm this hypothesis. Scientists from Marseilles are of the opposite opinion [23]. It is possible that the test should be repeated, assessing the intensity of pain after each tissue removal and comparing the obtained results with the time that has elapsed since the start of the procedure.

The work limitations include the fact that the study was conducted in a single center. The differences in the obtained results and assumptions of the study may result from the subjective, difficult to estimate assessment of the pain experienced by patients, the intensity of which is related to a different congenital threshold of pain sensitivity. Greater accuracy of the obtained results would allow a measurement of the level of perceived pain after each biopsy collection, during the procedure.

More limitations derive from how all of the biopsies were performed under local analgesia, from the lack of a control group (without analgesia) and from the patients’ different responses to the analgesia, each of which are results of the individualized conditions. It is also important that the procedure was not performed only by one operator. It should also be noted that the patient’s pain assessment was not performed after each sample collection.

## 5. Conclusions

The apex of the prostate is the most sensitive sector of the prostate due to its strong innervation. The prolonged duration of the fusion biopsy increases the severity of pain experienced by the patient. ISUP ≥ 2 and patients with low PSA levels more often indicated higher values on the pain rating scale. The age, size of the lesion and the number of samples taken do not correlate with a stronger feeling of pain. Similarly, PI-RADS, prostate zone anterior/posterior, and biopsy anxiety level showed no effect on the level of pain experienced during the procedure.

## Figures and Tables

**Figure 1 jpm-13-00431-f001:**
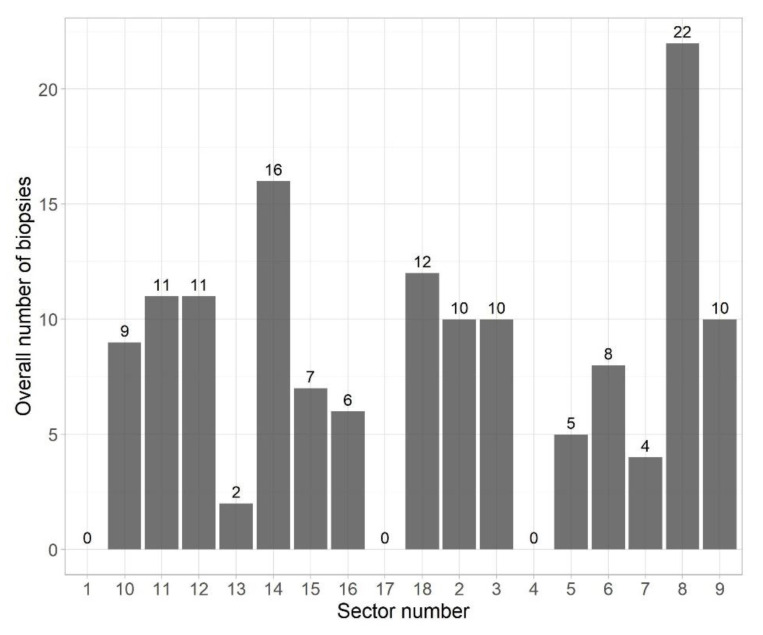
The overall number of biopsies taken from a given prostate sector.

**Figure 2 jpm-13-00431-f002:**
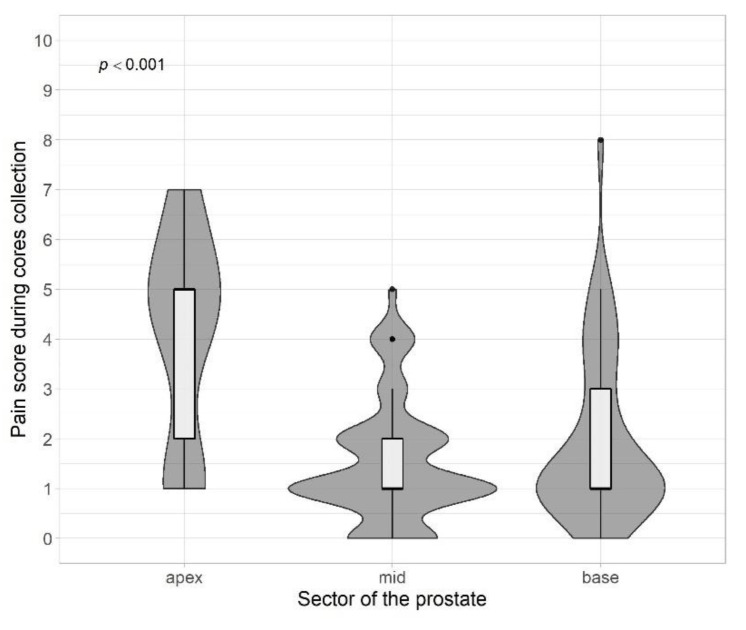
Pain score vs. target localization level.

**Figure 3 jpm-13-00431-f003:**
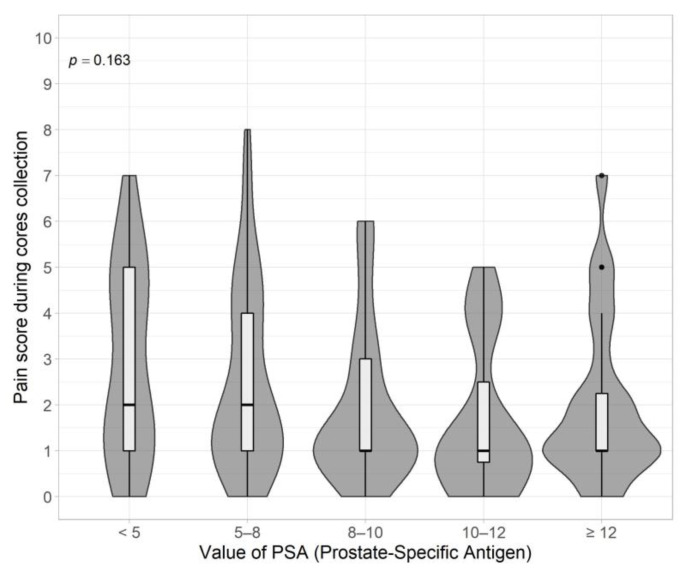
PSA value vs. pain experienced during biopsy.

**Figure 4 jpm-13-00431-f004:**
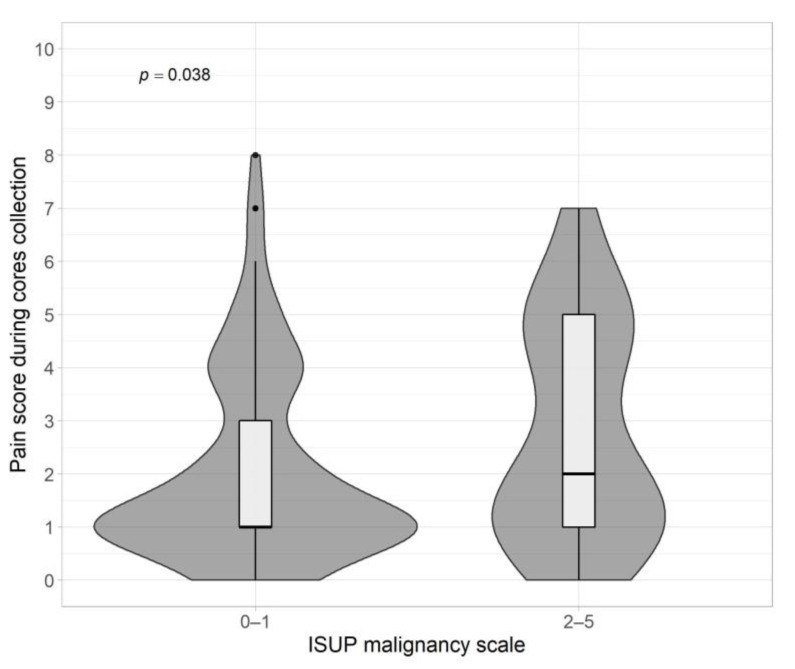
Pain score vs. ISUP malignancy scale.

**Table 1 jpm-13-00431-t001:** Tabular summary of data obtained from descriptive statistics.

Variable	Median	Mean	SD
Age (years)	66.4	65.6	7.4
Prostate volume (mL)	50.0	52.7	24.9
PSA (ng/mL)	7.6	9.7	10.1
PSAD (ng/mL^2^)	0.16	0.23	0.24
Biopsy time (min)	30.0	31.3	18.8
Target diameter (mm)	14.0	13.5	6.0
Number of cores	14.0	13.7	3.0

**Table 2 jpm-13-00431-t002:** The assessment of the relationship between perceived pain and clinical data.

Variable	*n*	Median	IQR	*p*
Prostate area				
posterior	81	2.00	1.00–4.00	0.099 ^a^
anterior	62	1.00	1.00–2.00	
Target localisation level				
Apex	34	5.00	2.00–5.00	<0.001 ^b^
Mid	66	1.00	1.00–2.00	
Base	43	1.00	1.00–3.00	
PSA				
<5	20	2.00	1.00–5.00	0.163 ^b^
5–8	54	2.00	1.00–4.00	
8–10	25	1.00	1.00–3.00	
10–12	16	1.00	0.75–2.50	
≥12	24	1.00	1.00–2.25	
ISUP				
0–1	102	1.00	1.00–3.00	0.038 ^a^
2–5	41	2.00	1.00–5.00	
PI-RADS				
2	9	2.00	1.00–5.00	0.42 ^b^
3	22	1.00	1.00–2.00	
4	69	1.00	1.00–4.00	
5	42	1.50	1.00–4.00	
Number of cores				
<10	12	1.00	0.00–4.25	0.74 ^b^
10–15	74	2.00	1.00–4.00	
>15	57	1.00	1.00–4.00	
Prostate volume				
<30	21	2.00	1.00–5.00	0.47 ^b^
30–60	59	2.00	1.00–4.00	
60–100	37	1.00	1.00–4.00	
≥100	5	1.00	1.00–1.00	

Note: ^a^ Mann–Whitney U test; ^b^ Kruskal–Wallis H test.

**Table 3 jpm-13-00431-t003:** The results of the analysis of correlation of particular variables on the level of pain experienced during the biopsy.

Variable	*p*-Value	r
Age	0.65	−0.04
Time of biopsy (affect the feeling of discomfort)	0.038	0.19
Target diameter	0.29	−0.10
PI-RADS	0.86	−0.02
ISUP (2–5)	0.29	0.09
PSA	0.046	−0.17
Number of cores	0.56	0.05
Prostate volume	0.22	−0.11
Anxiety	0.38	0.07

## Data Availability

The datasets used and/or analyzed during the current study available from the corresponding author on reasonable request.

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
