# Peer review of "The Severity of Pain in Prostate Biopsy Depends on the Biopsy Sector"

_jpm, 2023, doi:10.3390/jpm13030431_

Round 1
Reviewer 1 Report
Thanks for submitting your work on prostate biopsy sites and association with pain levels.
The following comments with regards to your study:
Language
“ peripastric nerve” should probably be “periprostatic”
Methodology
The biopsies were performed by more than one operator (number of operators not specifically stated). This may have a significant impact on the study results depending on the number of operators, their techniques, duration of the procedure and experience levels.
A better study design may have been to use patients as their own controls (to factor out the intra-individual pain threshold differences). In such a study design, the patient would score each site of biopsy in real time (rather than after the procedure).
It is not clear what the standard procedure for biopsy samples is (number, sites, approach, starting point, etc) and how this might be adapted based on your study findings.
Results
It is not clear what the explanation would be clinically, for an association between lower levels of s-PSA and higher levels of pain experienced.
The clinical implication of the study results is not clear.
Author Response
Author Response: We appreciate reviewers’ thoughtful comments which have substantially improved the content and clarity of this piece.
Comments and Suggestions for Authors
Thanks for submitting your work on prostate biopsy sites and association with pain levels.
Author Response: We would like to thank the reviewer for these nice and kind comments which encouraged us to improve as much as possible the manuscript according to the reviewer’s remarks in order to hopefully be accepted for publication in this prestigious journal.
The following comments with regards to your study:
Language
“ peripastric nerve” should probably be “periprostatic”
Author Response: Thank you very much! Indeed we made an error here! Corrected.
Methodology
The biopsies were performed by more than one operator (number of operators not specifically stated). This may have a significant impact on the study results depending on the number of operators, their techniques, duration of the procedure and experience levels.
Author Response: Again, the reviewer is right! Overall at our department and in the included studies the MRI fusion biopsies are being performed by two urologists (GR, MK) who got the same training, courses and overall, up to now have performed over 100 biopsies each. At the initiation of the studies the urologists were over their initial learning curve, however as the reviewer indicated physician-level factor could influence study results. Therefore we have added it as limitations (page 13). Missing information about operators has been added to the manuscript (page 4).
A better study design may have been to use patients as their own controls (to factor out the intra-individual pain threshold differences). In such a study design, the patient would score each site of biopsy in real time (rather than after the procedure).
Author Response: We appreciate this insightful point. We agree that in a perfect scenario the patients would score each biopsy site in real time, which sometimes could be challenging, however presumably feasible. Nevertheless, we have not done it at the time of the study protocol. We have added it now as limitation (page 13) and decided to update our studies designs on this topic for further publications to start assessment. We want to again thank the reviewer for this suggestion!
It is not clear what the standard procedure for biopsy samples is (number, sites, approach, starting point, etc) and how this might be adapted based on your study findings.
Author Response: The reviewer underlined an important point, we have added the protocol in the manuscript (page 3).
Initially, 4 samples were taken from suspicious lesions in MRI followed by 8-12 samples systematically collected the right and left lobes of the prostate. The number of samples depended on the assessment of the quality of the collected material by the operator during the biopsy. If the collected material was not suitable for histopathological evaluation, more specimens were collected. The biopsy was performed with transrectal access.
Results
It is not clear what the explanation would be clinically, for an association between lower levels of s-PSA and higher levels of pain experienced.
The clinical implication of the study results is not clear.
There is no known explanation for the relationship between low PSA and high pain. This requires further work and analysis. The clinical conclusion of this study may be that patients who undergo biopsies of lesions located in the anterior sectors of the prostate require additional anesthesia
We would like to thank the reviewer for these comments which encouraged us to make the best possible corrections to meet the criteria for publication in Journal of Personalized Medicine.
Reviewer 2 Report
the authors discuss an interesting issue on the hot topic of fusion transrectal biopsy under local anesthesia
Unfortunately, in this reviewer`s opinion, there are a number of problems that should be revised before eventual publication:
- peripastric nerve - maybe periprostatic nerves is more proper term
- in Material and Methods - a detailed description of operative technique is needed - equipment used, side-fire or end-fire probe, periprostatic block protocol etc.
- the main weakness in this study is that the pain intensity at different levels of the prostate (with focus on the apex) is assessed as a total for the whole study group, which is a surrogate marker. As the authors stated pain perception is very subjective and will be much more informative to assess pain after each puncture, which will give information for pain intensity score at different zones of the prostate in the same patient.
- the number of patient included is somewhat insufficient for such a multivariable analysis
- back and front of the prostate - unacceptable terminology for zonal anatomy of the prostate
- Figure 1 is unclear? not a percentage and not a overall number of cores?
- the statement that "prostate fusion biopsy is not a painful procedure" should be substantiated. The most significant argument for such statement is the high rate of willingness to undergo another biopsy if there is medical need - included in the authors` questionnaires, but not included in the results
- terms "analgesia" and "anesthesia" are not equivalent, they have significantly different meaning and should not be used as synonyms
Because of the aforementioned issues and some more minor ones, the manuscript is nor acceptable for publication in its current form in this reviewer`s opinion. It needs major revision before re-submission
Author Response
Author Response: We appreciate reviewers’ thoughtful comments which have substantially improved the content and clarity of this piece.
Comments and Suggestions for Authors
the authors discuss an interesting issue on the hot topic of fusion transrectal biopsy under local anesthesia
Unfortunately, in this reviewer`s opinion, there are a number of problems that should be revised before eventual publication:
- peripastric nerve - maybe periprostatic nerves is more proper term
Author Response: Thank you very much! We have corrected these errors in the manuscript.
- in Material and Methods - a detailed description of operative technique is needed - equipment used, side-fire or end-fire probe, periprostatic block protocol etc.
Author Response: A transrectal biopsy was performed using ultrasound BK Medical Flex Focus 400 -BioJet System (DK Technologies) It is a rigid fusion with a pneumatic arm to hold the ultrasound probe. Before the biopsy 11ml of Lignocaine gel was administered to the rectum, then, around the periprostatic nerve bundle, 5 ml of 1% lignocaine was injected on each side. Analgesia was performed with a Chiba 18G needle. Samples were taken by The Pro-Mag Ultra biopsy gun with an AIOU 14G 250mm needle. Missing information was added to the manuscript (page 3).
- the main weakness in this study is that the pain intensity at different levels of the prostate (with focus on the apex) is assessed as a total for the whole study group, which is a surrogate marker. As the authors stated pain perception is very subjective and will be much more informative to assess pain after each puncture, which will give information for pain intensity score at different zones of the prostate in the same patient.
We appreciate this insightful point. We agree that in a perfect scenario the patients would score each biopsy site in real time, which sometimes could be challenging, however presumably feasible. Nevertheless, we have not done it at the time of the study protocol. We have added it now as limitation (page 13) and decided to update our studies designs on this topic for further publications to start assessment. Our study provides insight into the topic, and now, due to comments from reviewers, we are planning a more detailed analysis. Once again, we would like to thank the reviewers for their knowledge and experience.
- the number of patient included is somewhat insufficient for such a multivariable analysis
Multivariate analysis were not performed, only univariate analysis, which include difference between group (Wilcoxon rank sum test or Kruskal–Wallis H test) and correlation.
- back and front of the prostate - unacceptable terminology for zonal anatomy of the prostate
I agree with this opinion. This is our oversight. An anterior and posterior zone is appropriate. Corrections have been made to the manuscript
- Figure 1 is unclear? not a percentage and not a overall number of cores?
Thank you for this opinion. Fig 1 has been improved. Additionally, a more detailed description has been added in the manuscript (page 5 ).
- the statement that "prostate fusion biopsy is not a painful procedure" should be substantiated. The most significant argument for such statement is the high rate of willingness to undergo another biopsy if there is medical need - included in the authors` questionnaires, but not included in the results
In fact, the manuscript lacks the data that allowed us to reach this conclusion. Results on the tolerability in biopsied patients are included in the manuscript (page 5). These data allowed us to draw a conclusion about the painlessness of this procedure.
|
Type of sensation |
|||||
|
Min. |
1st Qu. |
Median |
3rd Qu. |
Max. |
|
|
TOLERANCE |
0.00 |
0.00 |
1.00 |
2.00 |
10.00 |
- terms "analgesia" and "anesthesia" are not equivalent, they have significantly different meaning and should not be used as synonyms
Another oversight on our part. Of course, it's all about analgesia. Manuscript corrections have been made.
Because of the aforementioned issues and some more minor ones, the manuscript is nor acceptable for publication in its current form in this reviewer`s opinion. It needs major revision before re-submission
Author Response: We would like to thank the reviewer for these comments which encouraged us to make the best possible corrections to meet the criteria for publication in Journal of Personalized Medicine.
Round 2
Reviewer 1 Report
Thanks so much for addressing all of the issues raised in a detailed and positive way. Much appreciated!
Reviewer 2 Report
The authors had addressed all issues mentioned in the previous review, in this reviewer`s opinion this had resulted in improvement of manuscript.